# Interconnection between Microbiota–Gut–Brain Axis and Autism Spectrum Disorder Comparing Therapeutic Options: A Scoping Review

**DOI:** 10.3390/microorganisms11061477

**Published:** 2023-06-01

**Authors:** Angelo Michele Inchingolo, Assunta Patano, Fabio Piras, Antonio Mancini, Alessio Danilo Inchingolo, Gregorio Paduanelli, Francesco Inchingolo, Andrea Palermo, Gianna Dipalma, Giuseppina Malcangi

**Affiliations:** 1Department of Interdisciplinary Medicine, University of Bari “Aldo Moro”, 70124 Bari, Italy; angeloinchingolo@gmail.com (A.M.I.); assuntapatano@gmail.com (A.P.); dr.antonio.mancini@gmail.com (A.M.); ad.inchingolo@libero.it (A.D.I.); paduanelli@libero.it (G.P.); francesco.inchingolo@uniba.it (F.I.); giuseppinamalcangi@libero.it (G.M.); 2Implant Dentistry College of Medicine and Dentistry Birmingham, University of Birmingham, Birmingham B4 6BN, UK; andrea.palermo2004@libero.it

**Keywords:** autism spectrum disorder, microbiota, microbiota–gut–brain axis, gut health, nutrition, diet, probiotic, prebiotic, synbiotic, fecal microbiota transplantation, microbiota transfer therapy

## Abstract

Background: Autism spectrum disorder (ASD) is a group of neurodevelopmental illnesses characterized by difficulty in social communication, social interaction, and repetitive behaviors. These clinical diagnostic criteria can be seen in children as early as one year old and are commonly associated with long-term difficulties. ASD is connected with a higher frequency of various medical diseases such as gastrointestinal complaints, seizures, anxiety, interrupted sleep, and immunological dysfunction, in addition to the range of developmental abnormalities listed. Methods: From 1 January 2013 to 28 February 2023, we searched PubMed, Scopus and Web of Science for English-language papers that matched our topic. The following Boolean keywords were utilized in the search approach: “autism” AND “microbiota”. After deleting duplicates, a total of 2370 publications were found from the databases, yielding 1222 articles. (1148). Nine hundred and eighty-eight items were excluded after their titles and abstracts were scrutinized. The method resulted in the removal of 174 items for being off-topic. The final 18 articles for qualitative analysis are included in the evaluation. Conclusion: The findings of this extensive study revealed that probiotics, prebiotics, their combination as synbiotics, fecal microbiota transplantation, and microbiota transfer therapy may benefit ASD patients suffering from both gastrointestinal and central nervous system symptoms.

## 1. Introduction

Autism spectrum disorder (ASD) is a group of neurodevelopmental disorders characterized by a variety of deficits in behavioral areas such as social communication, social interaction, and repetitive activities [1]. These clinical diagnostic criteria can be seen in children as early as one year old and are often associated with lifetime difficulties [2]. ASD is associated with a higher prevalence of other medical conditions, such as gastrointestinal (GI) symptoms, seizures, anxiety, disordered sleep, and immune dysfunction, in addition to the spectrum of developmental impairments described; these conditions may have an impact on individual and family quality of life, as well as increase the cost and complexity of medical care [3,4,5]. GI problems, in particular diarrhea, constipation, and abdominal discomfort, are commonly reported in children with ASD, and growing data suggest that GI comorbidity may have downstream consequences on problematic behaviors in ASD [6]. A relationship between the gut and the brain has long been suspected, but research in recent years has begun to investigate the gut–brain interaction in ASD, leading to the identification of links between the gut microbiota (GM) and the pathophysiology of ASD [7]. Microorganisms (mostly bacteria, but also fungi, viruses, archaea, bacteriophages, and protozoa) that live in the lower GI tract, particularly the small intestine and colon, are referred to as the gut microbiome [7,8]. Individuals with ASD have altered gut bacteria profiles when compared to neurotypical controls, suggesting a potential role for GM in ASD [9,10,11]. The influence of GM on the GI system has been widely established, with GI motility, intestinal epithelial permeability, and mucus production all being influenced [12]. The severity of GI symptoms in ASD patients has been linked to derangements in the GM, such as during antibiotic administration. It was also discovered that if the antibiotics were withdrawn, the GI and behavioral problems resolved [13,14]. This offers new study opportunities for the function of GM-altering medicines such as probiotics as a possible treatment alternative. Recent research suggests that probiotics can help with a variety of psychological symptoms, including depression and anxiety [15,16]. The microbiota–gut–brain axis (MGBA) is thought to be a complex interaction between the brain and the GI tract [17]. The gut bacteria play a crucial role in controlling this gut–brain axis, and dysbiosis can have a deleterious impact not only on the GI tract but also on psychiatric symptoms [18,19]. Food restrictions and supplements have been studied in the therapy of ASD symptoms [20]. Apart from probiotics, the effect of prebiotics on the gut flora should not be overlooked. Nondigestible carbohydrates are one example. Exclusion diets and prebiotics were recently studied in children with ASD, with results showing substantial changes in GM composition and metabolism, as well as improvement in GI and behavioral symptoms [21,22]. A recent study found that transplanting gut microbial communities from ASD patients to wild-type germ-free mice induced classic autistic symptoms (increased repetitive behavior, decreased locomotion, and decreased communication compared to mice colonized with samples from typically developing controls) [23]. Treatment with microbial metabolites reduced (i.e., 5-aminovaleric acid) in the ASD microbiome; on the other hand, it regulated neuronal excitability in the prefrontal cortex (a regulator of social cognition), hence enhancing repetitive and social behaviors [23,24,25]. These findings provide additional evidence for a link between the GM and the area of the central nervous system (CNS) that underpins the pathology of ASD, providing a possible foundation for modulating the microbiota–gut–brain axis with microbial-based therapies to address ASD behavioral symptoms. Probiotics and prebiotics have received a lot of interest as possible therapy for ASD [26,27]. When a probiotic and a prebiotic are combined to provide health advantages, the combo is referred to as a complementary synbiotic (each component operates independently) or a synergistic synbiotic (prebiotic is selectively utilized by the co-administered live microorganisms) [28]. Fecal microbiota transplantation (FMT) is another treatment option being researched in ASD [29,30,31]. FMT may rebuild the recipient’s microbiota in a considerable and long-lasting way by injecting a solution of fecal matter from a healthy donor into the recipient’s digestive tract [32]. In 2019, the FDA awarded fast track status to an FMT therapy for the children with ASD [33]. In reference to this, one must also consider that microbiota transfer treatment (MTT), a treatment that includes antibiotics, an intestinal cleanse, a stomach acid suppressor, and FMT therapy, was developed. These therapies may influence ASD symptoms or progression via multiple GM-mediated immune, endocrine, and direct neural pathways (Figure 1) [7].

Locally, these actions may change the microbial ecology toward helpful bacteria and away from harmful bacteria [34]. Beneficial bacteria may boost the synthesis of microbial metabolites (for example, short-chain fatty acids (SCFAs)) and anti-inflammatory cytokines, which may improve intestinal barrier integrity and reduce intestinal and systemic inflammation [35]. Moreover, neuroactive metabolites such as SCFAs may have an influence on the CNS by modulating neuroplasticity, epigenetics, and gene expression [36]. The vagus nerve is a primary communication channel between the stomach and the brain that is activated in response to particular microorganisms. The stimulation of the vagus nerve, as carried out using *L. reuteri* therapy, may enhance oxytocin levels in the brain, positively altering behavioral elements of brain function [37]. Moreover, neurotransmitters and their precursors, such as gamma-aminobutyric acid (GABA), serotonin (5-hydroxytryptamine), tryptophan, glutamate, and dopamine, are produced by the microbiota [38,39]. Probiotics that stimulate inhibitory neurotransmission (e.g., greater GABA concentrations) may assist in restoring excitatory/inhibitory balance and hence correct the decreased social interaction associated with ASD [40,41]. Because there are few evidence-based therapy options for ASD, it is critical to investigate potential novel therapeutic targets for the social and behavioral symptoms. Preclinical studies on the GM offer intriguing new paths for behavior modification, and the rationale for testing gut microbial-based therapy for ASD grows stronger [42,43]. The goal of this scoping review is to provide an overview of the available data on the efficacy and safety of probiotic, prebiotic, synbiotic, and FMT therapies and MTT for the treatment of core and co-occurring behavioral symptoms in persons with autism spectrum disorder. The effect of these therapies on GI symptoms was also investigated. Several clinical trials have been filed with the goal of developing viable microbial-based therapeutics for ASD; this review provides an overview assessment of these trials as well as clarification of recent exploratory achievements.

## 2. Materials and Methods

### 2.1. Protocol and Registration

This review was conducted using the standards of the Preferred Reporting Items for Systematic Reviews and Meta-analysis (PRISMA) Extension for Scoping Reviews (PRISMA-ScR) [44].

### 2.2. Search Processing

We searched PubMed, Scopus and Web of Science with a constraint on English-language papers from 1 January 2013 through 28 February 2023 that matched our topic. The following Boolean keywords were utilized in the search strategy: “autism” AND “microbiota”. These terms were chosen because they best described the goal of our inquiry, which was to learn more about the interconnectedness between microbiota dysbiosis in patients with autism spectrum and whether the gut–diet–brain axis undergoing treatment through the use of prebiotics and/or probiotics, FMT, and MTT has positive neurological and gastrointestinal outcomes. The search indicators are listed below in Table 1.

### 2.3. Eligibility Criteria and Study Selection

We chose studies that looked at the effects of prebiotics, probiotics, synbiotics, FMT, and MTT on ASD. The selection method was divided into two stages: (1) title and abstract evaluation and (2) full text examination. Any article that met the following criteria was considered: (a) human intervention studies (clinical trials); (b) supplementation with probiotics, prebiotics, synbiotic combinations, FMT, or MTT; (c) studies assessing ASD; (d) treatment was compared to a placebo, no intervention, or other interventions; (e) English language full text; and (f) behavioral assessments were performed before and after the interventions using validated measures. Publications that did not include original data (e.g., meta-analyses, research procedures, conference abstracts, in vitro or animal studies) were excluded. The preliminary search’s titles and abstracts were retrieved and assessed for relevancy. For additional evaluation, full publications from relevant research were obtained. Two separate reviewers (F.P. and F.I.) evaluated the retrieved studies for inclusion using the criteria specified above, and disagreements were addressed by consensus.

### 2.4. Data Processing

Author differences over the article selection were discussed and resolved.

### 2.5. Data Extraction

A standardized form was used to capture data on research design and locations, population characteristics (e.g., sex, age, presence of comorbidities), type of intervention and comparison, baseline measurements, and reported results. Each study was also evaluated for its handling of missing data and effect measurements. For extraction accuracy, two reviewers (F.P. and F.I.) worked separately; divergences were resolved by consensus. Because of the substantial variability in the treatments and outcomes reported, meta-analysis was not possible; consequently, papers were synthesized qualitatively.

### 2.6. Data Analysis

For homogeneous research, the fixed effect model was used, while for heterogeneous studies, the random effect model was used. In all analyses, the effect size was calculated using the standardized difference of means.

### 2.7. PICOS Criteria

Table 2 depicts the PICOS (population, intervention, comparison, outcome, study design) criteria components, which include population, intervention, comparison, outcomes, and research design, as well as their use in this evaluation.

### 2.8. Study Evaluation

The article data were independently evaluated by the reviewers using a special electronic form designed according to the following categories: number of subjects, dose and type of intervention, study duration, type of study, age average of subjects, year of study, and main results.

## 3. Results

A total of 2370 publications were identified from the following databases: PubMed (720), Scopus (973), and Web of Science (677), which led to 1222 articles after removing duplicates (1148). Analysis of the title and abstract resulted in the exclusion of 988 articles. The writers successfully sought the remaining 234 papers for retrieval and evaluated their eligibility. The approach resulted in the exclusion of 174 articles for being off-topic. The evaluation includes the final 18 papers for qualitative analysis (Figure 2). The characteristics of the included studies are described in Table 3.

## 4. Discussion

We surveyed the original literature on therapeutic options targeting the GM for ASD in this scoping review, offering a resource to guide therapy based on evidence. There is mounting evidence that the GM may impact the onset and course of ASD. However, the lack of consistent knowledge implicit in the novelty of these considerations, as well as the ongoing lack of understanding of the complex microbial and metabolome distinctive signature in ASD patients, frequently translates into difficulties in microbiota-based therapy planning, which is typically performed on a basis of trial and error. This scoping review looked at 18 clinical studies that looked at the use of probiotics, prebiotics, probiotic/prebiotic compounds, FMT, or MTT in the treatment of core symptoms in juvenile ASD patients. The administration of these therapies had an effect on ASD symptoms in the fifteen RCTs. Moreover, the three non-RCT studies, on the other hand, imply that probiotics and prebiotics may modify behavior and GI symptoms in children with ASD. Prebiotics and the studied synbiotic formulations appear to be effective in selected ASD behaviors, although the extent of benefit is unknown, and there is less research that utilizes these techniques. Although clinical trial findings are limited, they urge additional investigation into FMT in ASD. This scoping review found that probiotic, prebiotic, synbiotic, FMT, or MTT supplementation improved different domains of ATEC score, including sociability, sensory or cognitive awareness, speech/communication/language, and health/physical/behavior, and there were improvements in behavioral symptoms in other studies that used different questionnaires (not ATEC) [29,45,46,47,48,50,52,54,55,57,58]. Concerning GI symptoms, all of the included trials found that probiotic, prebiotic, or symbiotic treatment improved the frequency of bowel movement-associated pain, diarrhea, constipation, and stool frequency [45]. It is difficult to determine which product is better for ASD symptoms among prebiotics, probiotics, or synbiotics because each study uses mostly one of these products, and even studies that use the same product differ in terms of dosage, treatment duration, or even the checklists they use to evaluate the results.

### 4.1. Probiotics/Prebiotics

Probiotics have the ability to mitigate gut dysbiosis, in some cases increasing the *Bacteroidetes/Firmicutes* ratio to that of healthy individuals; decreasing the growth of *Candida*, *Desulfovibrio*, and *Clostridia* species; and increasing beneficial bacteria such as *Lactobacilli* and *Enterococci* [61,62,63,64]. *Lactobacilli species* were also increased by prebiotic administration with galactooligosaccharide B-GOS^®^ [21,65]. MTT, on the other hand, appears to enhance *Desulfovibrio species* [29,56,60,61,66,67]. Probiotic investigation found a reduction in short-chain fatty acids (SCFAs), which are fermentation products of dietary carbohydrates generated by *Clostridium*, *Ruminococcaceae*, *Lachnospiraceae*, and *Desulfovibrio*, among others [50,60,68,69]. SCFA levels have been reported to be elevated in ASD [70,71,72,73]. However, with ketogenic diet (KD) adoption in ASD youngsters, which boosted SCFA-producing species, autistic core symptoms significantly improved but their significance in the etiopathogenesis of ASD remains unknown [74,75,76,77]. *L. reuteri*, for example, is an indigenous bacteria of the human GI tract that has been extensively researched, with accumulating evidence indicating its advantages as a probiotic [78]. In numerous mice models of ASD, *L. reuteri* was repeatedly demonstrated to produce OXT-dependent behavioral improvement [11,37,79,80,81]. It appears to have therapeutic promise in enhancing social and behavioral functioning in people with ASD. Nevertheless, only one experiment has been completed but not yet published that looked into the effectiveness of *L. reuteri* in ASD patients [50,82]. The bulk of probiotics utilized in research are from a small number of species, including *Lactobacillus* spp., *Limosilactobacillus* spp., and *Bifidobacterium* spp., and *Eubacterium coprostanoligenes* are not isolated from human GI tracts [58,83]. *Bifidobacteria* levels appear to be inversely associated via feedback interactions with *Desulfovibrio* and *Clostridium*, two of the most likely bacteria strains involved in ASD etiopathogenesis [84]. In this vein, our analysis suggests that when baseline counts are aberrant, enhancing *Bifidobacterial* populations (MTT, B-GOS^®^ supplementation) or reducing *Desulfovibrio* and *Clostridium* growth rates might be viable targets in microbiota-based ASD therapy [46,47,48,49,50,51,52]. It is critical to understand which gut commensals are associated with better symptoms in order to generate next-generation probiotics (also known as live biotherapeutic products) that evolve the features required to complete within the GI tract of ASD patients [85].

### 4.2. Alternative Medicines

It should be mentioned that alternative medicines such as Ayurveda, used in Dinesh’s study, also have positive implications on rebalancing the microbiota in ASD patients. A polyherbal formulation increased bifidobacterial abundance in the test group compared to the control group [53].

### 4.3. Synbiotics

Existing data on the effectiveness of probiotic, prebiotic, or synbiotic combinations in ASD is inconclusive and complicated by the fact that treatment regimens across studies are very varied, with varying formulations, doses, treatment periods, and administration procedures [86]. Increased efforts are recommended to focus on the effects of prebiotics in future clinical studies, since they may be safer, less expensive, affect a wide range of microorganisms, and be more widely accepted by all demographic groups [87]. Creating a synbiotic is more difficult. In an ideal world, a synbiotic would provide a health benefit greater than the sum of its individual components [88]. One published trial examining probiotic/prebiotic combinations in people with ASD did not systematically evaluate whether the combined products increase the ecological features and/or health effects of certain probiotic strains when compared to the probiotic alone [54,89]. Future research should look at other combinations and dosages, as well as comparing the ecological and functional aspects of each rationally chosen synbiotic to the substrate alone, the living microbes alone, and a control. At least one well-designed trial demonstrating a health advantage (complementary synbiotic) or both selective utilization of the substrate and a health benefit (synergistic synbiotic) in the target host has been carried out [28,54].

### 4.4. Dietary Supplements

Detailed dietary data should be included to assist in elucidating variables such as the influence of nondigestible carbohydrate consumption. In fact, the study by Bent et al. [55] discussed the beneficial effects of sulforaphane, which is a chemical produced when cruciferous vegetables such as broccoli, cauliflower, and broccoli sprouts are chewed. A component in these veggies called glucoraphanin interacts with a human enzyme called myrosinase to make sulforaphane. Through the intake of this product, it could be seen that urine metabolite changes were connected to oxidative stress, amino acid metabolism/gut microbiome metabolites, neurotransmitters, stress, and other hormones, while behavioral benefits have been associated with seven different chemical forms of sphingomyelin [55].

### 4.5. FMT and MTT

Fecal microbiota transplantation is an untargeted therapeutic for GM. Kang and colleagues created a modified FMT procedure called MTT for autistic youngsters [29]. This therapy alleviated ASD behavioral symptoms to some extent, with excellent tolerance indicated and improvements lasting 2 years after treatment ceased. The therapeutic relevance of these increases in behavioral assessment scores and improvement of GI disorders are confirmed in subsequent studies by Kang [57,60]: one of this two studies demonstrated that only p-cresol sulfate altered considerably following MTT. Significant associations have been made between p-cresol sulfate and *Desulfovibrio*, indicating that *Desulfovibrio* may have a role in p-cresol sulfate metabolism and the genesis of autism [60]. In a study by Turriziani et al., variation in p-cresol absorption appeared to contribute minimally, if at all, to behavioral alterations; in fact, a lowering of p-cresol was noted subsequent to PEG ingestion [45]. There are several other studies that support the beneficial effects of FMT and MTT, such as the Nirmalkar study, which found that the abundance of *Prevotella* and *Bifidobacterium* decreased over time (2 years), implying that a longer MTT treatment period or a booster after a certain amount of time may be required to retain these bacteria. MTT, in a similar manner, resulted in the normalization of numerous bacterial gene levels. Fascinatingly, microbial metabolic genes for folate biosynthesis, oxidative stress defense, and sulfur metabolism were dissimilar from those found in normally developing (TD) patients at ASD baseline but mirrored those found in TD and/or donors following MTT [56]. The positive effects of FMT can also be seen in a study by Li et al. [58]. In a meta-analysis of FMT in recurrent *Clostridium difficile* infection, Tariq et al. [90] discovered that the therapy was linked with decreased cure rates in randomized trials (67.7%) compared to open-label studies (82.7%; *p =* 0.001). To further study the causation between the GM and ASD symptoms, larger, randomized, double-blind trials comprising a matched control group of children with ASD who undergo an autologous transplant should be carried out. The FMT experiment discovered no difference in effectiveness between oral and rectal delivery [59]. Repeated FMT injections may be necessary to accomplish therapy goals. Depending on the mode of delivery, there are procedural hazards. For stool delivery, colonoscopy and oral routes have been demonstrated to be more successful than nasoduodenal tubes and enema [91]. It is also critical to anticipate potential issues with children with ASD’s willingness to receive FMT. However, FMT candidates should be examined in order to determine the most safe and effective FMT delivery mechanism in the setting of ASD. Continuing research into the mechanisms of action, the impact on the host’s immune response, and the refinement of the microbial inoculum may lead to a wider use of the treatment in the future. Furthermore, the safety profile of FMT in ASD is unknown, while the reported long-term experience of FMT for ASD patients in Kang and colleagues’ studies is reassuring. In the near future, further safety data will be provided by planned FMT studies that will conduct long-term monitoring and assessments. The reported FMT study subjects exhibited a variety of GI disorders, including diarrhea, constipation, and alternating diarrhea/constipation [29]. Current trials with bigger sample numbers involve more homogenous cohorts and may identify patients who are most likely to react [92]. In these research, adjunct therapies such as antibiotic pretreatment and colon cleansing were employed to reduce bacterial load and allow engraftment of potentially beneficial microbial taxa in the host [93]. Future research should focus on differences between patients within the same cohort as well as differences between study cohorts. Individuals’ illness presentations and levels of impairment might vary greatly. Individual disparities in responsiveness to intervention have been observed in previous studies into ASD treatments; treatment results may be impacted by biological characteristics such as age, linguistic ability, and autistic severity, as well as environmental factors (e.g., mother age and education) [94]. None of the included research investigated the possible impacts of traditional ASD medication therapies, such as proton pump inhibitors, as a confounding factor that might create a unique microbial profile [95].

Moreover, animal evidence shows that the microbiota’s modulatory CNS effects are sex-specific [96]; more female patients should be included in trials examining microbial-based therapy. Age is a crucial factor to consider in future ASD therapy research and should be adequately recorded in every intervention trial. Liu et al. [52] discovered that younger children who were treated with PS128 benefited more than older children. Previous research has found that a younger child’s age at the commencement of intervention is connected with favorable treatment results [94]. Probiotic, prebiotic, synbiotic interventions, FMT or MTT in early life with the goal of enhancing development in children at risk of ASD could become treatments of common use in the near future, as the GM shapes in infancy and any interventions in later years must be sustained for a long time, if not forever, to sustain the conferred benefit. Maintaining a healthy gut in infancy, which would most likely support healthy brain development throughout important developmental windows, may be more cost-effective and practicable than taking the supplements for extended periods of time later in life.

Based on the research presented in this review, it is theoretically possible to use multiple-strain combinations to investigate the attenuating effects of probiotics, prebiotics, synbiotics, FMT, and MTT on gut microbiota with the goal of reducing neurogastrointestinal symptoms in people with ASD. Similar research examining the effect of different strains on the GM can be conducted in order to create novel anti-ASD foods. Such analyses may reveal fresh and revolutionary pharmacological and food product pipelines with vast industrial uses. Some recent studies on the issue can serve as a good theoretical foundation for future study into GM regulation by assessing the impacts of these items. Future studies may need investigations into the manipulation of these pathogenic microbes for microbiota therapy reasons. In the near future, fresh starting cultures with more in vivo trials may emerge to support the hypothesis that microbiota therapies have more direct impacts on the inhibition of pathways and processes inside the human GM that predispose persons to ASD. This area of study is expected to significantly reduce the present ethical, cultural, and religious constraints inhibiting microbiota biotechnology research and the marketing of functional food components.

## 5. Conclusions

Several investigations in recent years have indicated qualitative and quantitative changes in the gut flora in a variety of neuropsychiatric illnesses, supporting the role of GM in the maintenance of physiological condition in the CNS. Within neurobehavioral disorders, it appears that at least a portion of ASD instances are linked to, and maybe reliant on, the health and wellbeing of the GM. The rising prevalence of ASD in recent years, combined with indications of a strong relationship between ASD and GI disorders, has sparked a specific interest in studying the reciprocal impacts between GM, brain, and microbiota under the so-called MGBA. The findings of this comprehensive analysis indicated that supplementation with probiotics, prebiotics, or their combination as synbiotics might be useful in alleviating both GI and CNS symptoms in ASD patients. It may also be argued that additional homogeneous trials using the same dosages and certain well-known probiotic or prebiotic products are required to apply the findings of these studies with more confidence. The research describes changes in GM composition in children with ASD that mostly consist of lower amounts of *Bifidobacterium* and higher levels of *Clostridium* spp. and *Desulfovibrio*. However, the current data do not allow for the definition of a distinct ASD profile. If dysbiosis is proven to be a precipitating factor in ASD, a variety of possible therapeutic options ranging from probiotics and prebiotics to FMT, MTT, and other nutritional techniques may be beneficial adjuvant therapy in these individuals. Furthermore, because dysbiosis contributes to a major proportion of ASD, identifying particular ASD endophenotypes would enable patient classification and targeted therapies. Addressing microbial processes might be the goal of the next ASD pharmaceutical therapy, which could assist in relieving the burden of this condition for the millions of individuals worldwide.

## Figures and Tables

**Figure 1 microorganisms-11-01477-f001:**
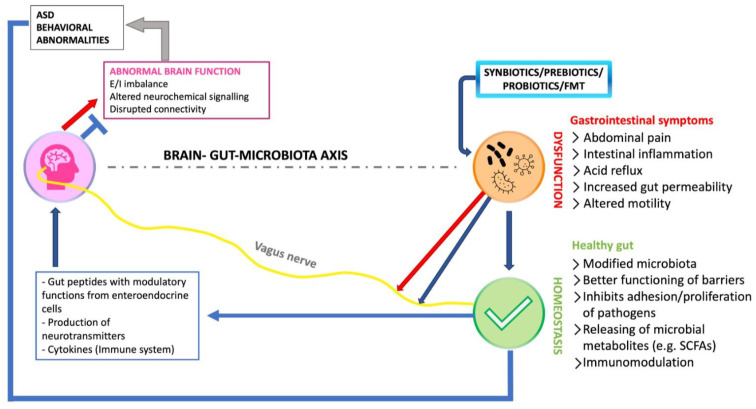
Manipulation of microbiota–gut–brain signals in autism spectrum disorder (ASD) using probiotics or fecal microbiota transplantation (FMT). The red arrows represent processes related with gut dysfunction and gut microbiota disturbances, whereas the blue arrows represent probiotic/prebiotic/FMT processes and effects. The vagus nerve (yellow) connects to enteric neurons and acts as a communication link between the gut and the brain.

**Figure 2 microorganisms-11-01477-f002:**
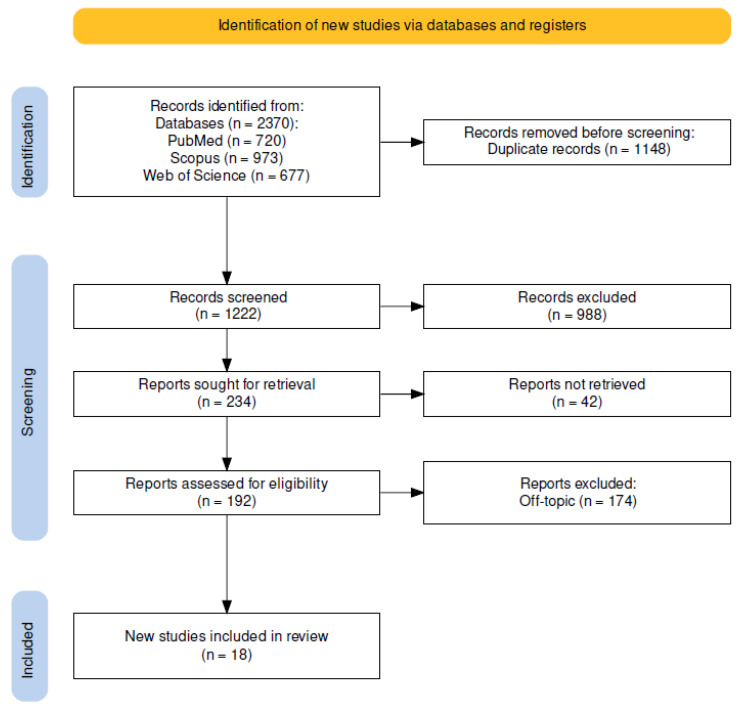
PRISMA flowchart diagram of the inclusion process.

**Table 1 microorganisms-11-01477-t001:** Database search indicators.

Article screening strategy	Database: Scopus, Web of Science, and Pubmed
	Keywords: A “AUTISM”; B “MICROBIOTA”
	Boolean variable: “AND”
	Timespan: 2013–2023
	Language: English

**Table 2 microorganisms-11-01477-t002:** PICOS criteria.

Criteria	Application in the present study.
Population	Subjects diagnosed with ASD according to DSM-5 criteria.
Intervention	Supplementation with probiotics or prebiotics or synbiotics, FMT therapy or MTT.
Comparisons	Comparing pre- and postintervention Bristol Stool Scale, Aberrant Behavior Checklist, Aberrant Behavior Checklist second edition, Autism Treatment Evaluation Checklist, Childhood Autism Rating Scale, Clinical Global Impression, Conners’ Parent Rating Scale—Revised and Social Responsiveness Scale, COoximetry, the fecal microbiome determined using 16 s rRNA sequencing, blood serum inflammatory markers, autoantibodies and oxytocin ratings in individuals after the study period.
Outcomes	Changes in behavioral test domains, in addition to the overall tests score. Changes in baseline and end in gastrointestinal and behavioral symptom measurements.
Study design	Clinical trials.

**Table 3 microorganisms-11-01477-t003:** Characteristics of the included studies.

Reference	Number of Subjects	Type of Intervention and Dose	Study Duration	Type of Study	Age Average of Subjects	Main Results
Probiotics/Prebiotics						
Turriziani et al., 2022 [45]	21	All children received gut mobilization using a conventional procedure that included once-daily oral injection of polyethylene glycol (PEG) at a dosage of 6.9 g/day. Urine was collected at each time point to measure p-cresol, stool quality was assessed by parental report using the Bristol Stool Scale, and parents completed the Repetitive Behavior Scale—Revised (RBS-R), Conners’ Parent Rating Scale—Revised (CPRS-R), and Social Responsiveness Scale (SRS), whereas ASD severity was measured using the Childhood Autism Rating Scale (CARS), which was administered by the same psychologist at all three time points for each subject.	26 weeks	Pre/post design	4.6 ± 1.7	The ingestion of PEG, which binds water molecules via hydrogen bonds, consistently enhanced bowel transit. Over a six-month period, gut mobilization was routinely followed by a gradual and substantial decrease in hyperactivity, anxiety, social interaction impairments, and stereotypic behaviors; hence, behavioral improvement was not only statistically significant, but also therapeutically useful. CARS overall score decreased by 7.7 points on average. Urinary p-cresol excretion patterns showed significant intra- and interindividual heterogeneity, with a rise at one month and a reduction at six months. The effect of PEG therapy on the microbiota is the most probable explanation for these alterations. PEG ingestion can influence microbiota composition via many routes, with distinct and potentially opposing consequences. Eight-to-twelve-point mean decreases in SRS, CARS, RBS, and CRPS scale scores. Urinary p-cresol levels showed unpredictable patterns that were not substantially associated to changes in behavioral characteristics. Total urine p-cresol, as determined in this study, is nearly exclusively composed of p-cresyl sulfate, with p-cresyl glucuronide and unconjugated free p-cresol accounting for little more than 5% on average. Variation in p-cresol absorption appears to contribute minimally, if at all, to these behavioral alterations.
Zhang et al., 2022 [46]	160	Probiotics—lyophilized powder mixtures containing *Lactobacillus*, *Bifidobacteria*, and *Streptococcus thermophilus* to be taken 10 billion colony units twice a day for three months, followed by a three-month washout period and a six-month follow-up. Maltodextrin is administered to the placebo group. All participants receive the Autism Treatment Evaluation Checklist (ATEC), CARS, social responsiveness scale second edition, children’s sleep health questionnaire, survey of food habits, GI assessment questionnaires, and the Bristol Stool Chart at each time point. At this time, the youngsters must refrain from taking antibiotics.	52 weeks	Parallel	N/A	Microbiota changes in children with ASD following probiotic administration may enhance microbiota balance and consequently ASD symptoms.
Sherman et al., 2022 [47]	35	For a total of 16 weeks, the probiotics group received oral probiotics PS128 (*Lactobacillus plantarum* PS128, a total of 6 × 10^10^ Colony Forming Units (CFU) per day), while the control group received oral placebo (microcrystalline cellulose). Post-hoc exploratory analysis. The outcomes of this study include the SRS, the Aberrant Behavior Checklist second edition (ABC-2), the Clinical Global Impression (CGI) scale, carboxyhemoglobin (SpCO) measured using COoximetry, the fecal microbiome as determined using 16 s rRNA sequencing, blood serum inflammatory markers, autoantibodies and oxytocin (OXT) as determined using ELISA.	16 weeks	Parallel	10.26 ± 4.78	Serum antitubulin, CaM kinase II, antidopamine receptor D1 (antiD1), and SpCO levels were found to be elevated in the bulk of ASD patients. In the therapy group, ASD intensity is associated with SpCO (baseline, R = 0.38, *p* = 0.029), antilysoganglioside GM1 (R = 0.83, *p* = 0.022), antitubulin (R = 0.69, *p* = 0.042), and antiD1 (R = 0.71, *p* = 0.045).
Guidetti et al., 2022 [48]	61	One sachet of the product every day containing 10 × 10^9^ CFU/active fluorescent units (AFU), 2.5 g of the probiotic mixture’s freeze-dried powder: *Limosilactobacillus fermentum* LF10 (DSM 19187), *Ligilactobacillus salivarius* LS03 (DSM 22776), *Lactiplantibacillus plantarum* LP01 (LMG P-21021), and a mixture of five strains of *Bifidobacterium longum* DLBL (probiotic formulation of Probiotical S.p.A., Novara, Italy) (DSM 22776), (LF10: 4 × 10^9^ CFU/AFU/dose; LS03, LP01, and DLBL mix: 2 × 10^9^ CFU/AFU/strain) The placebo group received sachets containing 2.5 g of powdered maltodextrin.	32 weeks	Crossover	4	Definite probiotics can lower the severity of behavioral and GI issues that often plague these individuals.
Kong et al., 2021 [49]	35	Probiotic— 6 × 10^10^ CFUs.	28 weeks	Parallel	9.85 ± 4.91	Nonsignificant improvement in total Autistic Behavior Checklist (ABC) score (*p* = 0.077).
Kong et al., 2020 [50]	60	For the first 12 weeks, all patients are randomly allocated to one of two groups: group A (30 subjects) receives oral *L. reuteri* probiotics (10^10^ CFU), whereas group B (30 subjects) receives an oral placebo. In the second stage, individuals in groups A and B continue to receive their respective oral *L. reuteri* probiotics or placebos as in stage 1. In addition, both groups are administered an intranasal OXT spray for an additional 12 weeks. Patients are started on 1 puff of 4 IU daily, and they also receive MRI training. Following one week, the dosage is increased to one puff in each nostril, twice a day (8 IU). Following the second week, the dose is increased to one puff in each nostril, twice daily (16 IU). After the third week, the dose is titrated up to the maximum amount of 24 IU daily, which is 2 puffs in each nostril in the morning and 1 puff per nostril in the afternoon. Even in younger individuals, a daily dose of 24 IU has been found to be safe and sufficient (ages 3–8 years old).	24 weeks	Parallel	N/A	(1) Variations in serum OXT levels, (2) variations in microbial relative abundance and diversity, and (3) variations in fecal short-chain fatty acid metabolites.
Arnold et al., 2019 [51]	13	Probiotic—900 billion.	19 weeks	Crossover	8.83 ± 2.80	Substantial enhancement in GI symptoms (*p* = 0.02).
Liu et al., 2019 [52]	80	Probiotic—*Lactobacillus plantarum* PS128- 3 × 10^10^ CFUs.	4 weeks	Parallel	10.01 ± 2.32	Certain behaviors improved.
Grimaldi et al., 2018 [21]	26	Prebiotic B-GOS^®^ (a galactooligosaccharide)-N/A.	10 weeks	Parallel	7.7	Improvement in GI disorders.
Alternative medicines						
Dinesh K.S. et al., 2022 [53]	60	Ayurveda polyherbal formulations (Rajanyadi Churna, Vilwadi Guilka) posology drug is administered with Lukewarm water thrice daily, 30 min before meals.	9 weeks	Parallel	N/A	After one month of evaluation, there was a substantial improvement in bifidobacterial abundance in the test group compared to the control group. After one month of follow-up in the test group, the higher quantity remained.
Synbiotics						
Sanctuary et al., 2019 [54]	11	Synbiotic and prebiotic comparison, 20 billion CFU/ day probiotic.	12 weeks	Cross-over	6.8 ± 2.4	Substantial improvement in GI problems as well as some behavioral symptoms.
Dietary supplements						
Bent et al., 2018 [55]	15	Sulforaphane (~2.5 μmol glucoraphanin (GR)/lb). Avmacol^®^, a sulforaphane-producing dietary supplement in tablet form, was used. Weight categories: 32–41 kg (6 tablets = 222 μmol GR/day), 41–50 kg (7 tablets = 259 μmol GR/day), 50–59 kg (8 tablets = 296 μmol GR/day), 59–68 kg (9 tablets = 333 μmol GR/day), 68–77 kg (10 tablets = 370 μmol GR/day), 77–86 kg (12 tablets = 444 μmol GR/day), 86–95 kg (13 tablets = 481 μmol GR/day), and 95–105 kg (15 tablets = 555 μmol GR/day).	12 weeks	Pre/post design	14.7	(1) Responders exhibited a 21.8-point decrease (improvement) in total ABC (*p* < 0.001) and a 20.2-point decrease in SRS (*p* < 0.001), compared to increases of 10 points in ABC (*p* = 0.001) and 8 points in SRS (*p* = 0.076) for non-responders. (2) Changes in urine metabolites were linked to oxidative stress, amino acid metabolism/gut microbiome metabolites, neurotransmitters, stress, and other hormones, while behavioral improvements were linked to seven unique chemical forms of sphingomyelin.
FMT or MTT						
Nirmalkar et al., 2022 [56]	18	Shotgun metagenomic study. Sequencing was performed using fecal DNA extracted from previous research [57].	18 weeks	Parallel	N/A	MTT changed the microbial composition of ASD individuals, causing many microbes to become depleted. *Prevotella*, *Bifidobacterium*, and the sulfur-reducer *Desulfovibrio* were among the beneficial bacteria that MTT also increased in abundance at the species level. However, *Prevotella’s* and *Bifidobacterium*’s abundances decreased over time (2 years), indicating that a longer MTT treatment period or a booster after a certain amount of time may be required to retain these bacteria. Similar to this, MTT also led to the normalization of many bacterial gene levels. Fascinatingly, microbial metabolic genes for folate biosynthesis, oxidative stress defense, and sulfur metabolism were distinct from typically developing (TD) patients at ASD baseline but resembled TD and/or donor levels after MTT (10 weeks, 2 yr).
Li et al., 2021 [58]	40	After the 4-week FMT treatment phase, there was an 8-week following observation period. FMT was administered to 27 children through freeze-dried pills, whereas colonoscopic FMT was administered to 13 children. Rectal route: a weekly dosage of 2 × 10^13^ CFU. Once a week, 50–100 mL per child. Oral route: 2 × 10^13^ CFU dosage, 8–16 capsules per child, once a week. The night before the transplant, the volunteers were administered 2 L of GOLYTELY (PEG). The same dose (about 2 × 10^14^ CFU per patient) was administered to both the oral capsule and the rectal administration groups once a week for four weeks.	12 weeks	Parallel	N/A	(1) In children with ASD, FMT was well tolerated and helpful in alleviating GI symptoms and autism-like behaviors. FMT appeared to cause the formation of a microbiota that differed greatly from the pre-FMT microbiota and was much more comparable to that of healthy donors and normally growing children. (2) Eubacterium coprostanoligenes that may be linked to treatment results.
Qureshi et al., 2020 [59]	38	Using ultrahigh-performance liquid chromatography–tandem mass spectroscopy, the researchers determined the content of 669 biochemical substances in the excrement of 18 ASD and 20 TD children. Following data were obtained from the ASD group during the 10-week MTT therapy and 8 weeks later. To describe changes in metabolites before, during, and after therapy, univariate and multivariate statistical analysis methods were used. The vancomycin dosage was tailored to each participant’s weight, starting at 40 mg/kg and rising to a limit of 2 g. After that, participants were exposed to one day of starvation and a stool cleanser (MoviPrep). The ASD cohort was split into two groups, each one following a different initial high dose (2.5 × 10^12^ cells/day) Standardized Human Gut Microbiota (SHGM) treatment. One MTT treatment consisted of a single dose administered rectally (n = 6) while the other involved doses administered orally on two days (n = 12). Both approaches were followed by an oral maintenance dose of a lower concentration SHGM (approximately 2.5 × 10^9^ cells), with therapy terminating 8 weeks after the first high dose. The procedure, however, changed differently for both sets of ASD youngsters who received SHGM rectally and waited one week before starting low dosage SHGM.	18 weeks	Parallel	N/A	Following MTT, observations show that the fecal metabolite patterns become more similar to those of the TD group. The median disparity between the ASD and TD groups was reduced by 82–88% for the panel metabolites, and 96% of the top fifty most discriminating individual metabolites reported more comparable values after therapy. As a result, these results are comparable, albeit less marked, to those obtained using plasma metabolites.
Kang et al., 2020 [60]	18	Ten weeks of MTT and an 8-week follow-up observation period (refers to Kang et al. [57]). To investigate whether metabolites were different in the ASD group before therapy, plasma and fecal samples were collected from children with ASD who had chronic GI difficulties (chronic constipation and/or diarrhea) vs. normally TD children who did not have GI problems.	10 weeks	Parallel	N/A	(1) Plasma metabolites: medium-chain fatty acids (caprylate and heptanoate), nicotinamide riboside, iminodiacetate, methylsuccinate, leucylglycine, and sarcosine were significantly lower at baseline and increased after MTT. (2) Fecal metabolites: Only p-cresol sulfate significantly changed after MTT. Significant correlations between p-cresol sulfate and *Desulfovibrio*, suggesting a potential role of *Desulfovibrio* in the metabolism of p-cresol sulfate and possible autism etiology.
Kang et al., 2019 [57]	18	MTT consisting of two-week vancomycin treatment followed by a bowel cleanse and then high dose FMT for 1–2 days and 7–8 weeks of daily maintenance doses. After this 10-week MTT treatment and an 8-week follow-up observation period (18 weeks in total), results were reassessed after 2 years of follow-up.	104 weeks	Pre/post design	N/A	Significant improvements both in GI and behavior symptoms.
Kang et al., 2017 [29]	18	MTT: - Vancomycin: 40 mg/kg by mouth per day, divided into three doses, not to exceed 2 mg per day. - Prilosec: 20 mg by mouth daily. - MoviPrep: Standard kit was used with half the dosage being administered at approximately 10 a.m. and the other half at 4 p.m. on day fifteen only, to cleanse the bowel of vancomycin and feces. The dosage varies proportionately based on the body mass. - Initial oral route: The dosage for the first 2 days (Day 16 and 17 only), 3 times a day for a total daily dose of 2.5 × 10^12^ cells/day. - Initial rectal route: 2.5 × 10^12^ cells, 1 time a day (Day 16 only). - Maintenance dose: 2.5 × 10^9^ cells, 1×/day by mouth.	18 weeks	Parallel	12 ± 5	Increase in variety andabundance of *Bifidobacterium*(×4), *Prevotella*, and *Desulfovibrio*;improvements in behavioral symptoms; significant improvements were seen in GI problems.

Abbreviations: ABC-2—Aberrant Behavior Checklist second edition; ATEC—Autism Treatment Evaluation Checklist; CARS—Childhood Autism Rating Scale; CFU—colony-forming units; CGI—Clinical Global Impression Scale; CPRS-R—Conners’ Parent Rating Scale—Revised; OXT—oxytocin; PEG—polyethylene glycol; RBS-R—Repetitive Behavior Scale—Revised; SpCO—carboxyhemoglobin; SRS—Social Responsiveness Scale; ABC—Aberrant Behavior Checklist; ASD—autism spectrum disorder; FMT—fecal microbiota transplantation; MTT—microbiota transfer therapy; PEG—polyethylene glycol; SHGM—Standardized Human Gut Microbiota; TD—typically developing.

## Data Availability

Not applicable.

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
