# Peer review of "Interconnection between Microbiota–Gut–Brain Axis and Autism Spectrum Disorder Comparing Therapeutic Options: A Scoping Review"

_microorganisms, 2023, doi:10.3390/microorganisms11061477_

Round 1

Reviewer 1 Report

The present study investigated the interconnection between microbiota-gut-brain axis and autism spectrum disorders (ASD) and compare the therapeutic options for the ASD. The results showed that probiotics, prebiotics, or their combination as synbiotics, fecal microbiota transplantation, and microbiota transfer therapy may benefit ASD patients suffering from both gastrointestinal and central nervous system symptoms. The manuscript was overall well written. Some minor points are listed as below.

1. Table 2 should be more explicit especially the type of intervention and dose and the main results.

2. The discussion part should be divided into several different topics and the limitations of the study should be discussed.

Moderate editing of English language are required.

Author Response

Dear reviewer, thank you for the suggestions provided. The answers are in the attached file. 

Sincerely

Reviewer 2 Report

1. 关于益生菌、益生元、合生元和 FMT 治疗自闭症谱系障碍的系统评价已有数项。类似的文章已经发表本研究并没有得出创新性的结论。因此,本研究不具有创新性。 

2.根据PRISMA 2020声明,系统评价的结果应包括:研究选择研究特征研究偏倚风险单项研究结果综合结果等。本文没有评估偏倚风险个体研究结果和综合结果。

3. 1和表2未做表缩略语未作解释。 纳入研究表2中未列出对照组或对照组。 

4. 在讨论部分:“关于胃肠道症状,所有纳入的试验都发现益生菌、益生元、共生素、FMT 或 MTT 治疗改善了排便相关疼痛、腹泻、便秘和排便频率。“和“关于胃肠道症状,所有纳入的试验都发现益生菌、益生元或共生素治疗改善了排便相关疼痛、腹泻、便秘和大便频率的频率”。和“此外,应包括详细的饮食数据以帮助阐明变量,例如摄入不易消化的碳水化合物的影响。”是对“此外,应包括详细的饮食数据以帮助阐明变量的重复”不易消化的碳水化合物摄入量”。此外,“生命早期的益生菌、益生元、合生元干预、FMT 或 MTT 以增强 ASD 风险儿童的发育为目标可能在不久的将来成为一个研究领域”的表达是模棱两可的。建议作者查看全文,删除或修改重复的地方。

Moderate editing of English language

Author Response

(The authors gave the same response as above.)

Reviewer 3 Report

This is a nice attempt to review how different therapeutic options may improve the gastrointestinal and behavioral symptoms of patients with ASD. I have some comments on how the method and results can be presented.

First, the search strategy is a bit too brief. Apart from “autism” and “microbiota”, what other keywords and abbreviations are being used? How do the keywords vary by databases? Usually authors would present those details in supplementary information and it’s good to have it for this paper too.

Second, how does study quality or risk of bias being assessed? Would the poor methodological quality of some included studies affect the validity of findings? This should be addressed by adding relevant assessment and discuss the findings later on.

Third, for records excluded, the reasons for exclusions should be explained in details instead of simply “off topics”. This applies for both abstract and full-text screening. In addition, there are too many papers with full-text not retrieved. Why is it so? Given only 18 papers are included for review, it raises the concern of limited representativeness because of not being able to get a large amount of papers.

Fourth, for the result section, it’s inappropriate to present everything in a big table. Authors should make good use of sub-headings to breakdown the efficacy of various treatment options.

Fifth, similar to result section, the discussion should also make good use of sub-headings to make the content more readable. Authors should also discuss the current treatment guidelines for patients with ASD, whether the present findings can inform the amendment in guideline? If not, what would be the next steps to improve the quality of evidence?

Lastly, it’s better to say “a systematic review of clinical trials” so readers can understand that these are evidence from intervention studies.

Nil

Author Response

(The authors gave the same response as above.)

Reviewer 4 Report

Work by Inchingolo et al. Concerning the Interconnection between Microbiota-Gut-Brain Axis and Autism Spectrum Disorder comparing therapeutic options: A Systematic Review, it is an interesting and well-written piece of literature. It requires a few minor changes to improve the readability and reception of the work, which I include below:

- I am asking you to standardize the notation of citations in the main text, because it seems that they have a completely different font;

- the introduction is extensively written and, admittedly, introduces the reader quite well to the issues discussed, but it could be slightly shorter;

- figure 1 is very interesting, but I would like to ask you to increase its quality and size to make it more readable;

- please reformat table 1 in accordance with the requirements of the journal (without shaded rows);

- Table 2 presents extremely important information, but it is quite difficult to follow in its current form. He proposes reformatting it to horizontal text orientation and combining, for example, the year with references, which will shorten the tables;

- additionally, abbreviations used in the table should be explained below;

- I would like to ask you to pay attention to the way of writing the names of microorganisms, which should be written in italics in most cases used in the text.

- the discussion could be diversified with a figure, because it is quite extensive.

Author Response

(The authors gave the same response as above.)

Round 2

Reviewer 3 Report

Authors have adequately addressed my comments, I have no further comments.

Nil